# La Jolla Virus: The Pathology and Transmission in Its Host *Drosophila suzukii*

**DOI:** 10.3390/v17030408

**Published:** 2025-03-13

**Authors:** Ibrahim Abdelhafiz, Tobias Kessel, Andreas Vilcinskas, Kwang-Zin Lee

**Affiliations:** 1Fraunhofer Institute for Molecular Biology and Applied Ecology, Branch of Bioresources, Ohlebergsweg 12, D-35392 Giessen, Germany; ibrahim.abdelhafiz@ime.fraunhofer.de (I.A.); andreas.vilcinskas@ime.fraunhofer.de (A.V.); 2Institute for Insect Biotechnology, Justus Liebig University of Giessen, Heinrich-Buff-Ring 26-32, D-35392 Giessen, Germany; tobias.kessel@ime.fraunhofer.de

**Keywords:** *Drosophila suzukii*, biological pest control, La Jolla virus, iflavirus, transmission, food intake

## Abstract

*Drosophila suzukii*, commonly known as spotted-wing drosophila, has emerged as a highly destructive pest in global fruit and wine production. The effectiveness of chemical control is significantly compromised by rapid resistance development and a limited range of insecticide options. Biological control presents a promising sustainable alternative. Our previous work suggested the La Jolla Virus (LJV) as a suitable candidate for the development of an insect virus-based control option. Here, we characterized the natural transmission and pathology of the virus. We tested various modes of horizontal transmission, including airborne, venereal and oral, and fecal routes. To understand LJV pathology in infected flies, we studied feeding behavior and demonstrated changes in food absorption compared to non-infected flies. We also investigated the impact on fecundity and egg-to-adult success rate. Altogether, these results collectively improve our understanding of LJV transmission in natural populations and the implication of infected flies in food ingestion and overall fitness.

## 1. Introduction

Since the beginning of its invasion more than a decade ago in California and Europe [1], *Drosophila suzukii* has spread across much of the globe [2,3], resulting in severe yield losses in many crops, such as cherries, berries, and grapes. Originally described in Japan and endemic to East Asia, this pest is now found on nearly every continent, causing annual damage in the millions [4,5]. Whereas most other Drosophila species feed and oviposit on decaying fruit, *D. suzukii* females have a preference to lay eggs in intact, ripening, and ripe fruit with their specialized serrated ovipositor. *D. suzukii* has an exceptionally broad range of host plants, targeting soft and ripening fruit across cultivated and wild environments [6,7]. With a high reproduction rate of ~400 eggs per female and a rapid life cycle of 8 days, the species demonstrates extraordinary adaptability. Its ability to tolerate diverse environmental factors and adjust reproductive strategies based on temperature and photoperiod makes it a highly resilient pest capable of causing significant agricultural devastation if left unchecked [8].

Traditional control methods for *D. suzukii* have relied heavily on conventional insecticides, including synthetic pyrethroids, organophosphates, spinosyns, and neonicotinoids. However, the application of synthetic insecticides comes with many disadvantages, including a high risk of insecticide resistance development [9] and harmful effect on the environment and non-target organisms [10,11,12,13]. Moreover, the *D. suzukii* larvae hatching inside the fruit are well protected from the environment and, due to the timing of application prior to harvesting, are difficult to treat. Thus, new and sustainable methods to control *D. suzukii* are urgently needed.

A promising strategy is the use of insect-specific viruses to control insect pests due to their generally high host specificity [13,14,15]. In previous studies, we identified a potential virus candidate, the La Jolla Virus (LJV), a positive-sense single-stranded RNA virus and member of the Iflaviridae family [16,17,18]. Initially discovered in a metagenomic approach in *Drosophila melanogaster* [16]. LJV was subsequently isolated from moribund *D. suzukii* specimens in Germany. Intrathoracic injection of LJV caused significant mortality among adult *D. suzukii* flies [17]. Similarly, oral administration of LJV also led to significant mortality in a concentration-dependent manner [19]. LJV demonstrates superior efficacy as an RNA virus against Drosophilids compared to Drosophila C Virus (DCV), particularly when administered orally. While DCV causes only 10–25% mortality in *Drosophila melanogaster* via oral infection, even at high titers [20,21], LJV significantly reduces the lifespan of infected flies, with all LJV-exposed flies succumbing earlier than controls.

Our research aimed to elucidate the transmission dynamics of LJV in *D. suzukii*. Building on previous studies that demonstrated oral infection and vertical transmission [19], we conducted experiments to further characterize horizontal, vertical, and potential vector-mediated transmission routes [22]. We investigated the impact of LJV epidemics on *D. suzukii* populations on lab scale and examined how viral infection affects the pest’s feeding behavior. These findings contribute to a more comprehensive understanding of LJV–host interactions, which is crucial for developing effective biocontrol strategies. A key question remains: Can a chronic or natural LJV infection significantly impact wild populations of *D. suzukii*? This study aims to address this critical question by investigating whether persistent viral infections can influence population dynamics, behavior, and the overall fitness of *D. suzukii* in natural settings, providing valuable insights into the potential of LJV as a biocontrol agent.

## 2. Materials and Methods

### 2.1. Drosophila suzukii Cultures Maintenance

The *D*. *suzukii* fly strain was derived from a laboratory colony established in the summer of 2012 in Ontario, Canada [23]. From this line, we produced one LJV negative and one LJV positive subline by adding orally infected flies to a negative line, as described previously [19]. The LJV negative line was maintained in a climate chamber (26 °C, 60% relative humidity, 12 h photoperiod) and checked regularly for LJV as described in 2.2. In contrast, the infected flies were kept in an incubator (model KBWF 240, Binder, Neckarsulm, Germany) under identical environmental conditions, with the addition of air circulation provided by a fan operating at 50% capacity (75 cm height × 60 cm width × 45 cm depth) and checked for LJV as described in 2.2 at the start of the performed experiments. All flies were maintained on a standard fly diet composed of 60 g/L cornmeal, 8 g/L soybean flour, 18 g/L baker’s yeast, 60 g/L malt, 22 g/L molasses, 6.25 mL/L propionic acid, and 2 × 0.8 g nipagin, which is poured in 50 mL drosophila vials. All experiments were carried out on flies aged 3–7 days post-eclosion. All experiments were conducted as 3 biological and technical replicates with the exception of the airborne transmission, which was carried out with 3 biological replicates and 4 technical replicates.

### 2.2. Extraction and Quantification of La Jolla Virus in Flies

RNA was extracted using a TRIzol reagent (Thermo Fisher Scientific, Waltham, MA, USA) from the collected flies, and 50 ng/µL aliquots were then prepared for quantification. The quantification was performed to assess the degree and progress of the infection in flies. Quantification was performed via StepOne real-time PCR system (Applied Biosystems, Waltham, MA, USA) with the Luna Universal Probe One-Step RT-qPCR Kit (New England Biolabs, Ipswich, MA, USA). Amplification was performed using the probes and primers described in Table 1. The real-time PCR was performed with the following steps: an initial heating to 55 °C for 10 min, followed by 95 °C for 1 min. This was followed by 40 cycles of 95 °C for 10 s for denaturation and 60 °C for 30 s for annealing, targeting positions 64–95 bp within the 128 bp product.

Viral infection levels were quantified using real-time PCR, with cycle threshold (Ct) values serving as the primary metric. Uninfected control flies consistently exhibited Ct values of approximately 30, while the highest infection levels observed corresponded to Ct values of around 10. This range of Ct values provided a robust measure of infection intensity across experimental conditions.

### 2.3. Transmission Assays

We arranged several experiments to determine if the transmission of the virus in a population of flies is airborne, venereal, oral, or fecal. For the airborne infection assay, we placed two vials of uninfected flies on the shelf at the top of the incubator, 5 cm from the infected fly culture, and two vials at the very bottom of the infected fly culture, 40 cm from the infected fly culture. Flies were regularly collected every 3 days (5 flies per interval and location). To test the hypothesis of venereal transmission of LJV within a population, different combinations of 10 virgin females were paired with 10 males. The combinations were as follows: As controls, we paired infected males and females (IVF × IM) and non-infected males and females (NIM × NIVF) as positive and negative baselines. The test conditions were Non-Infected Virgin Female and Infected Male (IM × NIVF), Infected Virgin Female and Non-Infected Male (NIM × IVF). As internal control, we set up pairings with the same-sex conditions to check for the requirement of the mating itself, or if factors such as close contact, including oral or fecal routes, are sufficient for transmission. The pairings went as follows: Non-Infected Virgin Female × Infected Virgin Female (NIVF × IVF) and Non-Infected Male × Infected Male (NIM × IM). All flies were allowed to pair for 3 days before collecting 5 of each sex. To differentiate the LJV negative flies from the infected, we cut off the wings of the infected flies. For the oral transmission, we placed 50 infected males on a vial of food and allowed them to feed for 3 days. Then we removed the flies, flooded the vial with CO_2_, and used a UV cross-linker (Stratagene Stratalinker 2400, La Jolla, CA, USA) to surface sterilize the vial in order to minimize fecal transmission. After this, 50 females and 50 males from the LJV negative flies were introduced and allowed to feed on the fly food. We then proceeded to collect 5 flies from both sexes every 3 days. For the fecal transmission, we placed a cotton ball saturated with 2 mL of 100 mM sucrose solution on parafilm in a completely empty drosophila vial. We then introduced 50 infected males and allowed them to feed and defecate on the walls of the vial for 3 days. After this, we removed the flies, the cotton ball, and the parafilm. Then, we thoroughly flooded the vial with CO_2_, added a fresh cotton ball on parafilm, introduced 50 LJV negative females and males to the vial, and allowed them to feed. Flies were collected every 3 days, and the cotton ball was replaced as necessary.

### 2.4. Egg to Adult Viability

To determine the influence of the virus on a fly population, we monitored the egg-to-adult viability of LJV-positive flies compared to LJV-negative flies. In short, we placed flies on grape juice agar plates (30% grape juice, 1% agar), transferring 100 eggs to a new agar plate. We then transferred the agar and placed it in a drosophila vial with food, then counted the larvae after 6 days, the number of pupae, and the number of emerged adults.

### 2.5. The Feeding Behavior of Infected Flies

To assess the impact of viral infection on feeding behavior, we employed the flyPAD (fly Proboscis and Activity Detector) system, an automated high-resolution behavioral monitoring tool that uses capacitive sensors to detect and analyze feeding behavior in Drosophila [24]. The flyPAD system allows for the extraction of various feeding metrics, including the number of activity bouts, the total duration of activity bouts, and the number of sips, all of which correlate with food intake. The system’s sensitivity enables the estimation of the volume of food consumed per sip, providing a comprehensive view of feeding dynamics in individual flies.

We adopted a slightly modified protocol based on previous work [24] The experiment consisted of three runs each for females and males, comparing 24 La Jolla Virus (LJV)-infected flies to 24 uninfected control flies per run. Prior to the experiment, flies were wet starved (females for 24 h and males for 18 h) to standardize hunger levels. Individual flies were then placed in the feeding arena of the flyPAD system. For each assay, we added 4 µL of either 5 mM or 50 mM sucrose solution to the reading electrode. Flies were allowed to feed for 1 h, during which their interactions with the food were continuously monitored by the flyPAD system. The system recorded multiple behavioral metrics, including the number of activity bouts (approaches to food), the total duration of activity bouts, and the number of sips. These parameters were used to estimate food intake and analyze feeding dynamics.

### 2.6. Statistical Analysis and Graph Design

All analyses were carried out via GraphPad Prism v9.1.2 for Windows (GraphPad Software, San Diego, CA, USA). For all the transmission experiments, a 2-way ANOVA test was run with multiple comparisons against an α of 0.05. The life stage experiment was performed via 2-way ANOVA, and the comparison was performed via Šídák’s multiple comparisons and compared to alphas of 0.05. The results and graphs of the behavior test are all calculated and plotted via the flyPAD program.

## 3. Results

### 3.1. Airborne Transmission

To investigate whether LJV is airborne transmitted, we raised non-infected flies close to LJV-infected flies in a closed environment, i.e., an incubator with constant temperature and humidity settings. Our findings revealed that infected flies exhibited high titers of LJV, with a baseline mean infection Ct value of 9.9. In contrast, the adjacent non-infected flies maintained their uninfected status throughout the observation period, from day 3 to day 12 post-exposure. The Ct values for these non-infected flies remained relatively stable and significantly higher than the established baseline threshold, confirming their uninfected state (Figure 1).

### 3.2. Venereal Transmission

Pairing experiments revealed distinct viral transmission patterns (Figure 2). Non-infected fly pairs maintained their uninfected status, as evidenced by high Ct values (approximately 30) in RT-qPCR analysis. In contrast, pairings involving infected flies resulted in high levels of LJV transmission, indicated by low Ct values (around 10). Notably, same-sex pairings showed transmission rates comparable to female–male pairings. This similarity does not exclude the possibility of venereal transmission but suggests that virus transmission occurs primarily through contact, likely via fecal or oral routes, rather than through venereal transmission.

### 3.3. The Transmission of the Virus Occurs by Oral and Fecal Contamination

The experimental outcomes for fecal and oral transmission routes showed striking similarities when assessing infection rates after several days of exposure (Figure 3A,B). Control flies maintained their non-infected status, as evidenced by high Ct values (approximately 30) in RT-qPCR analysis. In contrast, flies exposed to either feces or food from LJV-infected individuals exhibited significantly lower Ct values (approximately 20), indicating successful virus transmission through these routes. Notably, the transmission efficiency appeared to follow a time-dependent trend, with higher transmission rates observed after longer incubation periods. This temporal pattern suggests a gradual accumulation of viral particles in the exposed flies, potentially leading to more robust infections over time. These results collectively suggest the effectiveness of both fecal and oral routes in LJV transmission among *D. suzukii* populations, highlighting the virus’s capacity for efficient horizontal spread through environmental contamination.

### 3.4. Chronic LJV Infection Decreases Egg-to-Adult Success Rate

Our results demonstrate a significant reduction in the percentage of emerged adults due to the impact of the virus (Figure 4). While all eggs successfully hatched, differences began to emerge at the larval stage, particularly when larvae reached the L3 stage. The virus seems to disrupt normal larval growth, resulting in a gradually widening gap in survival rates when compared to uninfected controls. Although significant differences were observed across all developmental stages, the most pronounced effect was seen during adult emergence. At this stage, the virus caused a substantial decline of around 40% in the number of adults successfully completing pupation and emerging.

### 3.5. LJV Affects the Feeding Behavior of Female Flies

We employed the flyPAD (fly Proboscis and Activity Detector) system to assess whether LJV infection alters the feeding behavior of *D. suzukii*. Our results reveal significant changes in the feeding patterns of infected females compared to uninfected controls.

The feeding behavior of non-infected and infected flies was evaluated for both genders (Figure 5A,B) using the cumulative duration of activity bouts. This metric, representing the total time engaged in physical activity over multiple bouts during a one-hour period (x-axis in Figure 5A,B), represents in the flyPAD system the sum of all individual activity bout durations recorded for a fly during the observation period. It is calculated in the flyPAD system by summing all individual activity bout durations for each fly. This measure effectively captures the full spectrum of fly movement patterns, from brief exploratory actions to extended locomotion periods, making it valuable for assessing overall activity levels. While female flies exhibited a notable difference, males showed no significant variation.

In contrast, the cumulative number of feeding bursts represents the total count of discrete feeding events recorded by the flyPAD system over a specified observation period. In Drosophila studies using flyPAD technology, a feeding burst is typically defined as a short, intense period of proboscis extension and food interaction. This metric provides a quantitative measure of feeding frequency, offering insights into the flies’ overall food consumption patterns and feeding behavior. By summing these individual feeding events, the total feeding activity can be assessed, which may be influenced by factors such as the LJV infection status. The cumulative number of feeding bursts serves as an indicator of feeding drive and can be used to compare feeding behaviors across different experimental conditions. In our investigation, no significant difference (*p*-value 0.068 for females; *p*-value 0.12 for males) was observed between noninfected and infected flies, suggesting no substantial variation in feeding uptake (Figure 6A,B).

## 4. Discussion

We studied the transmission routes and pathologies of the La Jolla Virus (LJV) in its host, *Drosophila suzukii*. Although insect viruses typically do not spread through the air, it is worth noting that certain human respiratory viruses, such as coronaviruses, influenza viruses, and rhinoviruses [25,26], can be transmitted via airborne routes. Our research suggests that airborne transmission does not occur, at least under the specific laboratory conditions we used. The study’s experimental findings validate that LJV primarily spreads through the oral–fecal route, with venereal transmission likely playing a secondary role. This finding is consistent with prior studies on iflaviruses, which have demonstrated diverse transmission strategies, including horizontal (oral–fecal) and vertical routes, depending on the host species and environmental conditions. For example, *Euscelidius variegatus iflavirus* 1 (EVV-1) primarily spreads via fecal–oral transmission, as observed in *Nilaparvata lugens* honeydew virus-1 and Nora Virus in other insect species [27,28]. Deformed Wing Virus (DWV), an Iflaviridae family member akin to LJV, illustrates the complex transmission dynamics and infection effects seen in insect RNA viruses [29]. The transmission routes and infection outcomes of DWV are among the most extensively studied within the Iflaviridae family, offering valuable insights into the ecology of these viruses. The entoparasitic mite *Varroa destructor* serves as a crucial vector for DWV, substantially influencing its virulence. The concurrent presence of DWV and Varroa mites results in symptoms like pupal mortality and the characteristic deformed wings of worker bees [30]. Although LJV appears to infect without a vector, similar symptoms, such as pupal death, can be observed in *D. suzukii* LJV infections. A common effect is the severely shortened adult life span in an acute DWV infection. In mite-free conditions, DWV develops into a persistent, covert (chronic-asymptomatic) infection. In contrast, LJV seems to establish a persistent and chronic symptomatic in the absence of a vector, significantly impacting population dynamics by reducing offspring numbers and the fitness of the emerging flies. This outcome is very important since an application of LJV as a biocontrol agent would also be able to spread chronically within a population and maintain their virulence.

Next, we explored the possibility of venereal transmission of the virus. Venereal transmission of arboviruses has been documented in several cases [31,32,33,34]. To test this hypothesis, we paired infected flies with their mates and monitored them for infections. All flies paired with infected partners showed initial signs of infection, which was a promising indicator. However, infection was also observed in same-sex pairings, suggesting potential environmental transmission rather than direct sexual contact. To confirm sexual transmission conclusively, future studies should involve dissecting and examining sexual organs for viral presence. If confirmed, sexual transmission could be a valuable mechanism, especially when combined with other techniques, such as the Sterile Insect Technique (SIT). Sterile infected males that have been sterilized with eco-friendly methods [35], could be released as viral vectors to disseminate the infection throughout the population. While venereal transmission cannot be excluded for LJV, its significance appears to be secondary to the oral–fecal route. Studies on other insect-specific viruses, such as Culex flavivirus (CxFV), have shown that venereal transmission plays a minor role compared to vertical and horizontal routes [36]. Similarly, recent research on medfly associated iflaviruses indicates that while vertical transmission via females is predominant, male-mediated venereal transmission is possible but less efficient [37]. Studies comparing iflaviruses have revealed diverse transmission methods within this viral family. While some iflaviruses, like the *Antheraea pernyi* Vomit Disease virus [38] and *Spodoptera exigua* iflavirus [39], primarily transmit vertically through eggs or larvae, this study did not focus on vertical transmission. However, its potential role in LJV epidemiology deserves further exploration.

The predominance of oral–fecal transmission has significant implications for understanding host–virus interactions and population-level dynamics. Insects with gregarious feeding behaviors or those inhabiting densely populated environments are particularly susceptible to rapid viral spread via contaminated food or substrates. The systemic nature of infection ensures that all life stages contribute to environmental contamination. This continuous viral shedding increases the chances of sustained viral circulation within host populations, even in the absence of overt disease symptoms. Additionally, investigating how environmental factors like temperature or humidity affect fecal–oral transmission efficiency could provide valuable context for understanding seasonal variations in infection rates. Future research should also investigate potential interactions between LJV and other pathogens or microbiota within the host gut. As demonstrated by SeIV1’s interaction with baculoviruses [40], such associations may modulate viral infectivity and persistence.

We subsequently evaluated the complete life cycle of both uninfected and infected flies to determine their egg-to-adult survival rate. This investigation builds upon a previous study [19], which demonstrated that oral infection with LJV negatively affects fly eclosion. Our aim was to verify these findings and explore the potential impact on a chronically infected fly population, thereby demonstrating the potential of LJV as a biocontrol agent. Our findings revealed that LJV significantly impairs the adult emergence rate in fly populations. This is in contrast to the picorna-like Nora virus, which can establish infections in laboratory fly strains and persist for several years without causing any significant pathological effects [41]. The persistent presence of LJV in the environment could serve as a continuous control measure, which, when combined with other complementary methods, may lead to population suppression. The results indicate that the virus has its most significant effect during the later developmental stages, possibly affecting essential physiological or metabolic functions required for pupation and adult eclosion. This underscores the virus’s capacity to influence population dynamics by decreasing reproductive success and adult survival rates in *D. suzukii*, even in cases of long-term, chronic infection within the population. Nevertheless, we are unable to distinguish between potential maternal influences, where the chronic infection may have resulted in reduced yolk or protein deposition in the eggs, and developmental effects, where the infected larvae’s food consumption is impacted. Additional research is necessary to address this question.

Finally, we examined the virus’s effect on fly feeding behavior. Our findings indicate that the virus has a more pronounced impact on female flies, with infected females showing reduced feeding-relevant activities compared to their uninfected counterparts. This suggests a reduced drive or motivation to seek out food sources among infected females. This observation is particularly relevant in natural environments where food is likely to be more dispersed and limited, potentially affecting fly survival rates. Interestingly, male flies do not exhibit the same response to the virus, showing no significant differences in feeding behavior.

Our study reveals that while infected flies spend more time locating food, their overall food consumption remains unaffected. Although the statistical difference was marginal (*p*-value slightly exceeding the 0.05 α threshold), these findings could have significant implications in natural settings, especially in areas with scarce food resources. In field conditions with regular maintenance and fewer hiding spots for flies, we anticipate observable effects, particularly given the infected flies’ reduced motivation to search for food. Our research also highlights opportunities for further investigation into how the virus influences *D. suzukii*’s feeding behavior. Future studies could explore whether the virus affects specific receptors and, if so, which ones. Additionally, research could examine if the virus impacts fly mobility and whether different food types alter the foraging behavior of infected flies.

Viruses offer a promising alternative due to their specificity and effectiveness, and LJV has shown significant antagonistic effects against *D. suzukii*. However, like other insect viruses, the application of LJV presents its own set of challenges. Previous research has demonstrated that, once infected, the virus can kill the insect within 5–8 days [19]. This relatively slow action may limit its immediate impact in controlling infestations. To address these challenges, studying the transmission dynamics of LJV within *D. suzukii* populations is a prerequisite for a controlled release in natural environments. Conducting field trials is crucial for gaining insights in natural settings, as laboratory conditions often fail to capture the complexities of real-world environments. Furthermore, developing viral formulations to improve stability and delivery, or employing strategies like continuously infecting food sources with viral particles, could strengthen the practical application of LJV as a biocontrol agent. The mass production of LJV for biological pest control would require the development of cost-effective, large-scale fermentation processes involving cell culture-based methods. Engineering could potentially enhance the speed and effectiveness of LJV’s lethal capabilities. Nevertheless, the public generally disfavors genetically modified organisms (GMOs). As a result, approaches that utilize natural selection processes are more widely accepted as alternatives. Additionally, care must be taken to minimize off-target effects, as members of the Iflaviridae [42] are known to be antagonists of bees, raising ecological concerns. Combining LJV with other management techniques, such as SIT, could maximize its efficacy and reduce the likelihood of resistance developing in the *D. suzukii* population.

In summary, our study highlights the critical importance of the oral–fecal route as the primary mechanism for transmitting LJV and the long-term effects of chronic infection, including feeding behavior change on a population level. These findings align with the existing literature on iflaviruses and provide a foundation for further exploration into host–virus dynamics and ecological implications. Understanding these transmission pathways is essential for developing targeted strategies to manage infections caused by LJV and related viruses in insect populations.

## Figures and Tables

**Figure 1 viruses-17-00408-f001:**
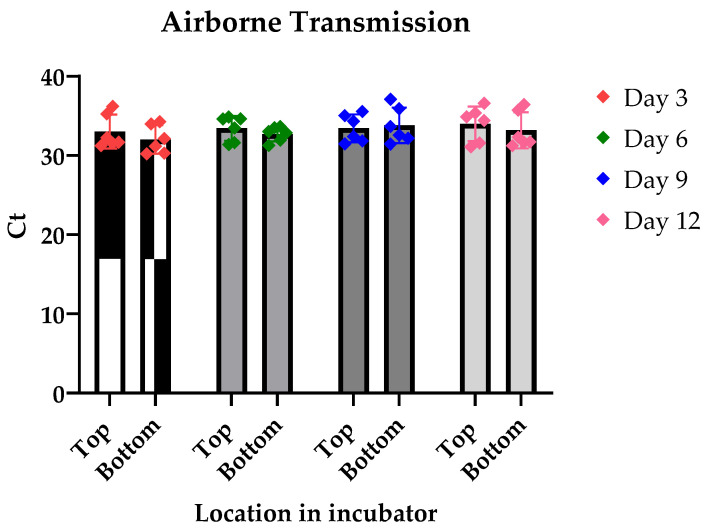
The results of the real-time PCR after several days of the flies being in the vicinity of an infected fly culture. The graph demonstrates no significant change in the Ct of the extracted samples, concluding there was no infection. Results show no significant difference in the infection levels as the days went on. SD of day followed by top/bottom: Day 3: 2.136/1.788; Day 6: 1.567/0.925; Day 9: 1.755/2.241; Day 12: 2.183/2.277. The error bars represent the standard error mean (SEM). *p*-value = 0.38 alpha = 0.05.

**Figure 2 viruses-17-00408-f002:**
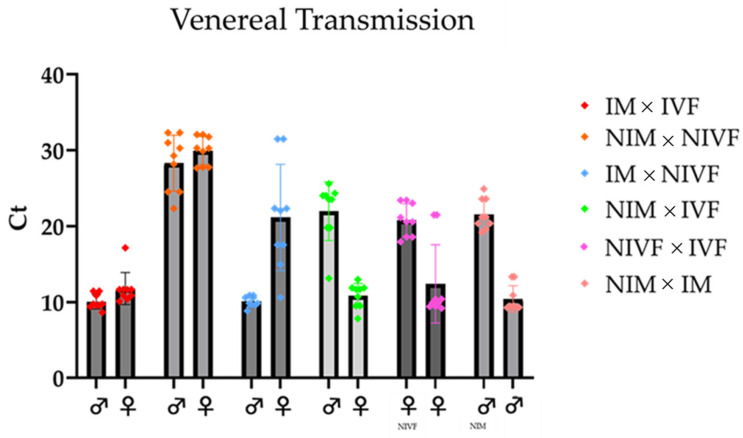
The infection levels after the flies were allowed to mate for 3 days. The first two sets are the positive (IM × IVF, red dots) and negative controls (NIM × NIVF, orange dots), followed by the different combinations tested. All infection levels are significant compared to the negative control (NIVF + NIM) alpha = 0.05.

**Figure 3 viruses-17-00408-f003:**
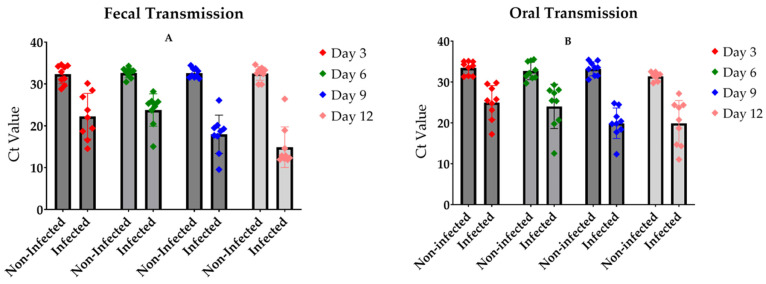
The infection levels in the flies ((**A**) Fecal transmission; (**B**) Oral Transmission) after several days of infection. The figure shows a steady infection rate in both experiments. Infected flies showed a significant difference to the non-infected control (*p*-value < 0.0001 in **A**,**B**).

**Figure 4 viruses-17-00408-f004:**
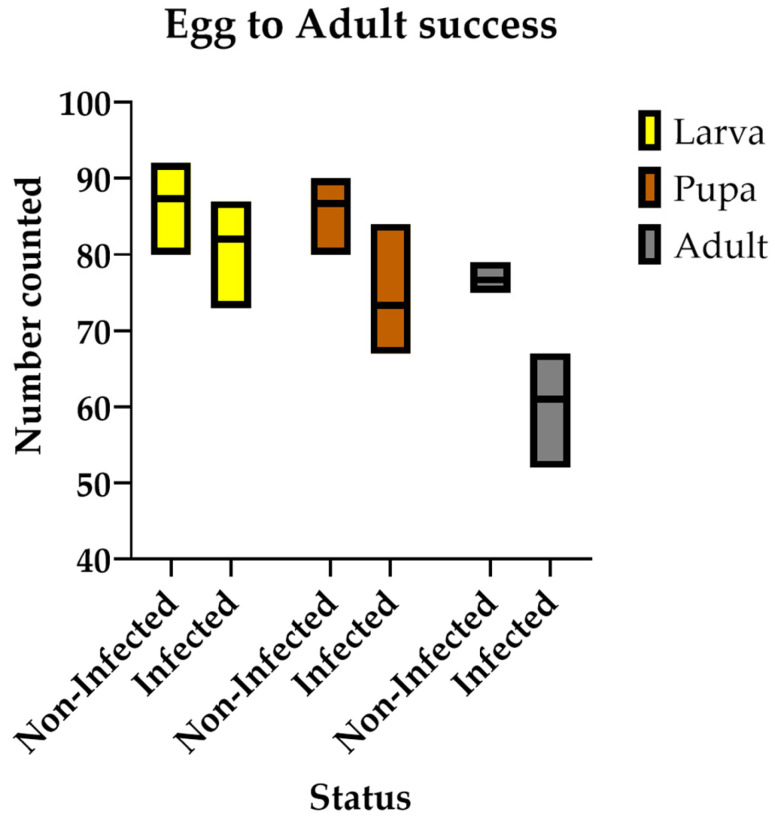
Egg-to-adult success rate of non-infected versus infected flies for larvae (yellow bars), pupa (orange bars), and adults (gray bars). All differences are significant when α = 0.05. SE of difference = 3.274.

**Figure 5 viruses-17-00408-f005:**
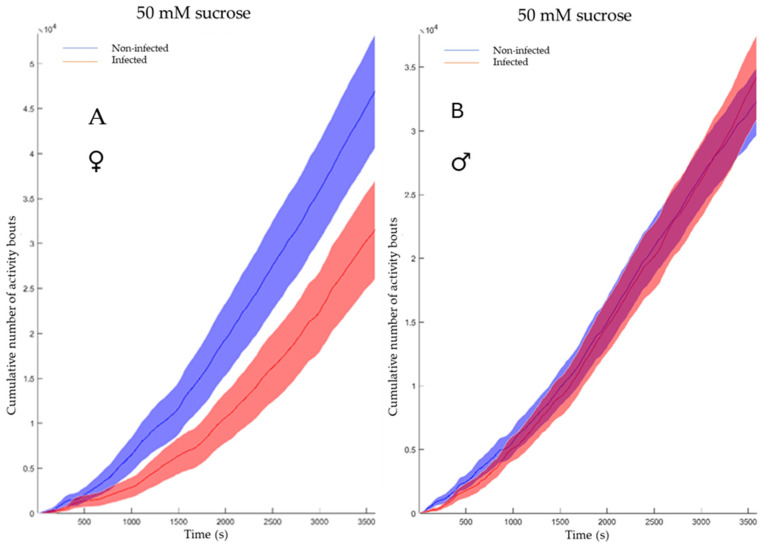
Cumulative duration of activity bouts. (**A**) The feeding activity of the non-infected females (blue) vs. the infected females (red). The software conducted a Mann–Whitney test and shows a *p*-value of 0.036. (**B**) The feeding activity of the non-infected males (blue) vs. the infected males (red) The software conducted a Mann–Whitney test and shows a *p*-value of 0.93.

**Figure 6 viruses-17-00408-f006:**
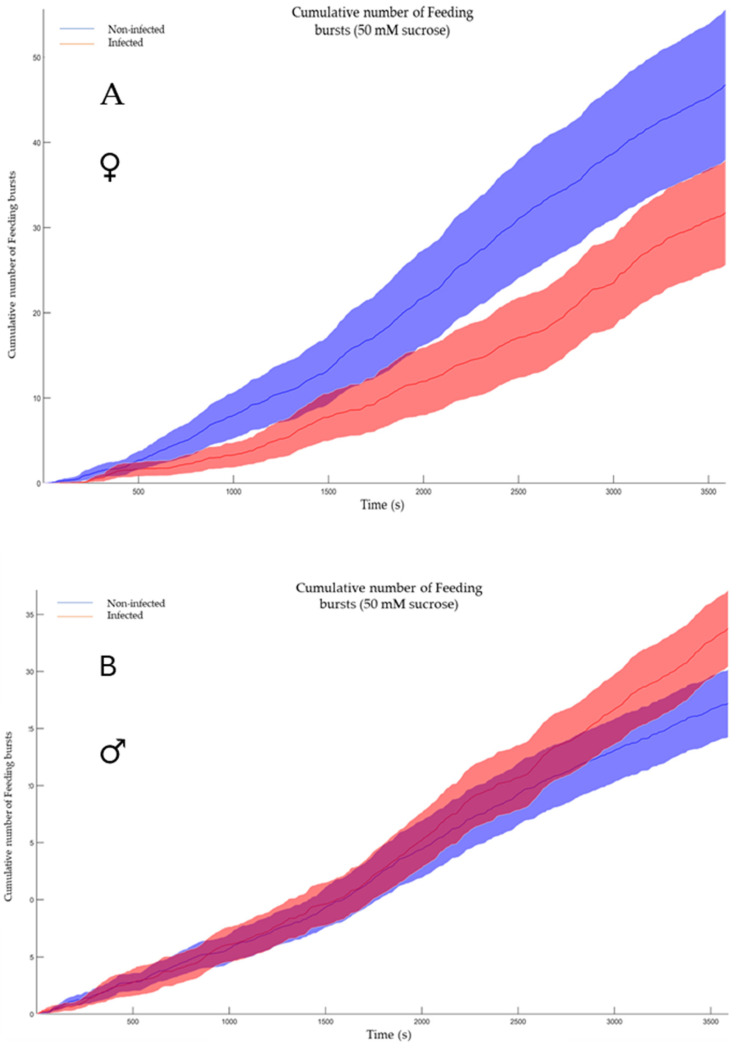
Shows the cumulative number of feeding bursts. (**A**) The feeding activity of the non-infected females (blue) vs. the infected females (red). The software conducted a Mann–Whitney test and shows a *p*-value of 0.068. (**B**) The feeding activity of the non-infected males (blue) vs. the infected males (red) The software conducted a Mann–Whitney test and shows a *p*-value of 0.12.

**Table 1 viruses-17-00408-t001:** Shows the sequence of the primers and probes used in this study.

Description	Sequence	Product Length (bp)
LJV specific probe *	5′-ACTCGGCGTTATCGTTACAACCGCACATATC-3′	
LJV forward primer	5′-CAACACGTTGTGCTGCCTGA-3′	128
LJV reverse primer	5′-TCCATCCAAACTCCACCTCC-3′	128

* Labeled on 5′ with fluorescent reporter dye FAM, on 3′ with fluorescent quencher TAMRA. The probe is at position 64–95 bp within the 128 bp product.

## Data Availability

The data presented in this study are available on request from the corresponding author.

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
