# Peer review of "La Jolla Virus: The Pathology and Transmission in Its Host Drosophila suzukii"

_viruses, 2025, doi:10.3390/v17030408_

Round 1

Reviewer 1 Report

Comments and Suggestions for Authors

Abdelhafiz et al further the investigation of Drosophila suzukii La Jolla Virus in this work. The virus was previously described to negatively impact D. suzukii, in particular, among drosophilid flies: virus-infected flies exhibited shorter lifespan than uninfected, raising the possibility that LJV could negatively impact D. suzukii fitness, including reduced fecundity, pupariation, and eclosion success. The virus can chronically infect lab populations, raising questions as to transmission route within infected populations, as well. They tested these phenomena through lifespan assays, activity and feeding assays, and infection/transmission route assays. Their primary findings were an absence in transmission in (largely) stale air environments, mechanical transmission through fecal-oral route, reduced pupariation and eclosion rates in infected lines, and reduced total and feeding activity in infected female populations, but no change in males, relative to uninfected.

The primary concern from this study is the appearance that the bulk of the work centered around particularly the transmission and life history impact studies, were carried out a single time. This may only require clearer communication, but from my reading there is note only of the number of flies used and not the number of independent replications – for example, lines 123-128, the authors note they placed 50 infected males in vial then transferred 100 flies into the tube and proceeded to collect them later. This represents an n=1, not n=100 (or 50). This contrasts to the behavioral analyses, where it clearly states that the work was performed three times.

Beyond this, the following minor errors and issues should be addressed:

Line 66: authors claim to examine impact of virus epidemics on populations, going on in line 70 (and a few other places) to claim they’re being tested in natural settings. However, the impact of an epidemic hasn’t been tested here, rather, they’re testing the impact of virus infection on life history and potential transmission modalities in a lab, not natural, environment.

Lines 77-79: how was the infected line established? How was it validated including how often?

Lines 99-100: table information appears incomplete

Lines 108-112: the airborne transmission assay has serious deficiencies, I believe. First, air movement within the climate chamber (lines 79-80) was not communicated suggesting that the air movement is minimal and not representative of wind-potentiated movement. Secondly, a  positive control for wind transmission would be useful both to validate that air movement is occurring and secondly to verify that the plugs used in vials allow for particulate passage – this last is probable given LJV’s size but it would still be good to validate and provide a positive control for your experiment design.

Lines 123-128: the authors should provide rationale for UV treatment and demonstrate that UV-light is insufficient to modify virus infectivity (in short, it is unclear why the authors UV treated here and while I predicted such treatment might inactivate any residual LJV, and this was not borne out, other modifications beyond deactivation of the virus by UV cannot be ruled out)

Lines 129-135: how does this experiment design discriminate fecal as compared to oral or simply contact? I do not see a large difference here between potential for flies to defecate on the walls and uninfected to contact (129-135) and for them to defecate on the food and for uninfected to contact (123-128).

Line 136: the authors do not test fecundity (egg/offspring counts) rather than count hatching, pupariation and eclosion success

137-142: not a lot of detail here but suggests the “LJV Line” was used, could there have been selection or other impact on the chronic infection, which could alter results? In other words, how do we discriminate between maternal (maybe the mother didn’t load as much yolk or proteins into eggs in the infected group) and ontogenic (maybe the larvae do not eat as much) effects.

Figure 2 caption: what does the bar represent – mean or median? What about the variance?

Lines 193-195 (and in discussion): the authors claim that the results indicate that venereal is a secondary transmission route as they see similar Ct values in homo-sexual tubes. This is subjective! It could be that transmission modality changes depending on fly population density and sex composition and would need more specific testing to determine the weighting.

Fig 3: the requirement for color makes this harder to read and utilize if one prints in out on a B&W. It seems like it would be both clearer and easier for the reader if the X-axis labels were more explicit rather than relying on the legend.

Figs 4-5: the authors do not comment on something I found intriguing which is the consistency of the Ct values in infected individuals in these graphs: there is no significant change d3-d12, although there is a trend toward reduction. What is the meaning of this? Does this look like other replication trends in newly infected individuals? Or, does it represent that a significant portion of the individuals tested in the “infected” are in fact uninfected – that is, maybe the reducing mean Ct (as per Fig 2 is it mean, or median and what are the errors?) represents an increasingly binary split between some individuals with a very hot infection and some that just aren’t infected?

Line 221: “While all eggs successfully hatched” – was this determined?

Lines 226-227: the data in fig 5 suggest there is a decline of ~25-33% in adult eclosion, not 40%, as they go from UI 80% to Inf 60% (25% decline). Even if the authors are calculating a decline from ~90% UI pupariating to ~60% Inf eclosing that is still “just” a 33% decline.

Lines 334-336: no reference is given for the SeIV1 x baculovirus interaction

Line 340: throughout the manuscript the authors use the term “emergence”, as in “negatively affects fly emergence”. I would encourage the authors to use the specific correct term “eclosion” to refer to adult emergence from the pupa (or more correctly in drosophilids, the puparium) as this will remove confusion as to the stage of which they are discussing.

Lines 354-371: I disagree with the interpretation that infected flies are spending “more time to find food” – what is this based on? The authors’ relevant data are in figs 6 & 7, where 6 shows total activity (movement, feeding attempts, etc) and the infected females exhibit significantly less activity; fig. 7 shows reduced active feeding activity in infected females. The easiest interpretation here is that infected females are “more lethargic” ie exhibit reduced activity overall. This matters as other studies of D. melanogaster, for example, demonstrates feeding shifts to compensate for metabolic insufficiencies associated with pathogens (as well as self-medicating behaviors). The activity results further support this as males have different metabolic profiles and energetics to females – maybe females are more negatively affected by infection than males, which may in fact change their behaviors. This is a possibly very interesting aspect of the work! If the authors noted sex of eclosing adults (Fig 5) that might be insightful, as might the number of eggs laid by uninfected vs infected females.

Author Response

Reviewer 1:

Abdelhafiz et al further the investigation of Drosophila suzukii La Jolla Virus in this work. The virus was previously described to negatively impact D. suzukii, in particular, among drosophilid flies: virus-infected flies exhibited shorter lifespan than uninfected, raising the possibility that LJV could negatively impact D. suzukii fitness, including reduced fecundity, pupariation, and eclosion success. The virus can chronically infect lab populations, raising questions as to transmission route within infected populations, as well. They tested these phenomena through lifespan assays, activity and feeding assays, and infection/transmission route assays. Their primary findings were an absence in transmission in (largely) stale air environments, mechanical transmission through fecal-oral route, reduced pupariation and eclosion rates in infected lines, and reduced total and feeding activity in infected female populations, but no change in males, relative to uninfected.

The primary concern from this study is the appearance that the bulk of the work centered around particularly the transmission and life history impact studies, were carried out a single time. This may only require clearer communication, but from my reading there is note only of the number of flies used and not the number of independent replications – for example, lines 123-128, the authors note they placed 50 infected males in vial then transferred 100 flies into the tube and proceeded to collect them later. This represents an n=1, not n=100 (or 50). This contrasts to the behavioral analyses, where it clearly states that the work was performed three times.

Thank you for pointing this out. All experiments were conducted in three biological replicates, each biological replicate was performed in three technical replicates except for the airborne transmission. Therefore, we have added a paragraph in line 90-93: “All experiments were conducted as 3 biological and technical replicates with the exception of the airborne transmission which was carried out with 3 biological replicates and 4 technical replicates (two on the top of the incubator and two on the bottom).”

Beyond this, the following minor errors and issues should be addressed:

Line 66: authors claim to examine impact of virus epidemics on populations, going on in line 70 (and a few other places) to claim they’re being tested in natural settings. However, the impact of an epidemic hasn’t been tested here, rather, they’re testing the impact of virus infection on life history and potential transmission modalities in a lab, not natural, environment.

Agree. We have accordingly modified our statement to “D. suzukii populations on lab-scale” on page 2, line 67, to emphasize this point.

Lines 77-79: how was the infected line established? How was it validated including how often?

We added orally infected flies as described in Linscheid et al., 2022 in the clean stock and validated the infection status at the start of the performed experiments. We revised this accordingly at line 79-80. We validated the LJV status by qPCR, described in section 2.2 at the start of the experiments. We added this information “and checked for LJV as described in 2.2 at the start of the performed experiments” in line 84-85.

Lines 99-100: table information appears incomplete

Thank you for pointing this out! We have completed the Figure legend of Table 1. To “ * Labeled on 5’ with fluorescent reporter dye FAM, on 3’ with fluorescent quencher TAMRA. The probe is at position 64-95 bp within the 128 bp product.” At line 99-100.

Lines 108-112: the airborne transmission assay has serious deficiencies, I believe. First, air movement within the climate chamber (lines 79-80) was not communicated suggesting that the air movement is minimal and not representative of wind-potentiated movement. Secondly, a positive control for wind transmission would be useful both to validate that air movement is occurring and secondly to verify that the plugs used in vials allow for particulate passage – this last is probable given LJV’s size but it would still be good to validate and provide a positive control for your experiment design.

We agree that the airborne transmission assay lacks clear description, since air circulation was provided but not explicitly stated in the methods section. We have now changed the paragraph (line 82-85) accordingly to “In contrast, the infected flies were kept in an incubator (model KBWF 240, Binder, Neckarsulm, Germany) under identical environmental conditions, with the addition of air circulation provided by a fan operating at 50% capacity (75 cm height x 60 cm width x 45cm depth)”. We thank reviewer 1 for the idea of a positive control, however the technical implementation would be out of the scope of our experiment.

Lines 123-128: the authors should provide rationale for UV treatment and demonstrate that UV-light is insufficient to modify virus infectivity (in short, it is unclear why the authors UV treated here and while I predicted such treatment might inactivate any residual LJV, and this was not borne out, other modifications beyond deactivation of the virus by UV cannot be ruled out)

Our aim was to use UV treatment to surface sterilize the walls of the vial to minimize the possibility for an infection via feces. We changed to “…to surface sterilize the vial in order to minimize fecal transmission” (line 130-131).

Lines 129-135: how does this experiment design discriminate fecal as compared to oral or simply contact? I do not see a large difference here between potential for flies to defecate on the walls and uninfected to contact (129-135) and for them to defecate on the food and for uninfected to contact (123-128).

The oral transmission test relies on the food, since the vial surfaces, where most of the defecation occurs, has been surface sterilized by UV, and infected flies were removed, that no direct contact between infected and uninfected flies existed. The fecal transmission test is void of food, which is replaced by a cotton ball with the minimal food source sucrose. Both, infected flies and cotton ball were removed prior to addition of uninfected flies, removing food sources for oral infection.

Line 136: the authors do not test fecundity (egg/offspring counts) rather than count hatching, pupariation and eclosion success

Agree. We have accordingly removed fecundity in this section (see paragraph 2.4., lines 140-141).

137-142: not a lot of detail here but suggests the “LJV Line” was used, could there have been selection or other impact on the chronic infection, which could alter results? In other words, how do we discriminate between maternal (maybe the mother didn’t load as much yolk or proteins into eggs in the infected group) and ontogenic (maybe the larvae do not eat as much) effects.

Thank you for the comment. Indeed, we haven’t discriminated between the two effects and therefore did not mention it in this section. We have added a paragraph in the discussion section (at line 354-358): “Nevertheless, we are unable to distinguish between potential maternal influences, where the chronic infection may have resulted in reduced yolk or protein deposition in the eggs, and developmental effects, where the infected larvae's food consumption is impacted. Additional research is necessary to address this question.”

Figure 2 caption: what does the bar represent – mean or median? What about the variance?

The bars represent the standard error mean (SEM), instead of variance, we added now standard deviation (SD) to quantify variability among replicates. The figure legend reads as “SD of day followed by top/bottom: Day 3: 2.136/1.788; Day 6: 1.567/0.925; Day 9: 1.755/2.241; Day 12: 2.183/2.277. The error bars represent the standard error mean“ (lines194-196).

Lines 193-195 (and in discussion): the authors claim that the results indicate that venereal is a secondary transmission route as they see similar Ct values in homo-sexual tubes. This is subjective! It could be that transmission modality changes depending on fly population density and sex composition and would need more specific testing to determine the weighting.

To our interpretation of the results, the veneral transmission together with the results of the oral-fecal route are indicating a more likely oral-fecal route of transmission, as exemplified by most other studied iflaviruses. We toned the claim down by adding: “This similarity does not exclude the possibility of venereal transmission, but suggests that virus transmission occurs primarily through contact, likely via fecal or oral routes, rather than through venereal transmission.” (line 201-202).

Fig 3: the requirement for color makes this harder to read and utilize if one prints in out on a B&W. It seems like it would be both clearer and easier for the reader if the X-axis labels were more explicit rather than relying on the legend.

Thank you for your suggestion. Since most of the reader are relying on PDF version of the manuscript, we would like to keep the color representation of the figure.

Figs 4-5: the authors do not comment on something I found intriguing which is the consistency of the Ct values in infected individuals in these graphs: there is no significant change d3-d12, although there is a trend toward reduction. What is the meaning of this? Does this look like other replication trends in newly infected individuals? Or, does it represent that a significant portion of the individuals tested in the “infected” are in fact uninfected – that is, maybe the reducing mean Ct (as per Fig 2 is it mean, or median and what are the errors?) represents an increasingly binary split between some individuals with a very hot infection and some that just aren’t infected?

Thank you for your comment. The consistency in Ct values observed in infected flies from days 3 to 12 is believed to represent the baseline for the infectious state. Any variations or trends towards decrease are considered to reflect only the individual flies' infection levels.

Line 221: “While all eggs successfully hatched” – was this determined?

Yes, we documented the hatching rate, which was 100 %, so 100/100.

Lines 226-227: the data in fig 5 suggest there is a decline of ~25-33% in adult eclosion, not 40%, as they go from UI 80% to Inf 60% (25% decline). Even if the authors are calculating a decline from ~90% UI pupariating to ~60% Inf eclosing that is still “just” a 33% decline.

Agreed. We rectified our typo accordingly to “around 33% (line 234).”

Lines 334-336: no reference is given for the SeIV1 x baculovirus interaction

We have now added the reference for the SeIV1 x baculovirus interaction (Jakubowska et al., 2016).

Line 340: throughout the manuscript the authors use the term “emergence”, as in “negatively affects fly emergence”. I would encourage the authors to use the specific correct term “eclosion” to refer to adult emergence from the pupa (or more correctly in drosophilids, the puparium) as this will remove confusion as to the stage of which they are discussing.

Agree. We have replaced “emergence” with the term “eclosion” in discussion section, line 346, 349 and 356, except in the sentence “…the most pronounced effect was seen during adult emergence” (line 234), where it seems to fit better than the term eclosion.

Lines 354-371: I disagree with the interpretation that infected flies are spending “more time to find food” – what is this based on? The authors’ relevant data are in figs 6 & 7, where 6 shows total activity (movement, feeding attempts, etc) and the infected females exhibit significantly less activity; fig. 7 shows reduced active feeding activity in infected females. The easiest interpretation here is that infected females are “more lethargic” ie exhibit reduced activity overall. This matters as other studies of D. melanogaster, for example, demonstrates feeding shifts to compensate for metabolic insufficiencies associated with pathogens (as well as self-medicating behaviors). The activity results further support this as males have different metabolic profiles and energetics to females – maybe females are more negatively affected by infection than males, which may in fact change their behaviors. This is a possibly very interesting aspect of the work! If the authors noted sex of eclosing adults (Fig 5) that might be insightful, as might the number of eggs laid by uninfected vs infected females.

Thank you for your comment. We have revised this part and the paragraph reads now “infected females showing reduced feeding relevant activities compared to their uninfected counterparts.” In line 367-368.

Reviewer 2 Report

Comments and Suggestions for Authors

Drosophila suzukii is emerging as a viable threat to the fruit and wine industries globally. This fruit fly species is difficult to control for several reasons, including resistance to insecticides and is location and foraging inside fruits.  The authors conducted routine and basic work to build upon a previous study that suggested the La Jolla virus (LJV) could be an alternative biocontrol agent for this pest. In this study the authors characterized the transmission and pathology of LJV by investigating the modes of transmission (airborne, venereal and oral, and fecal routes); feeding habits; and fecundity. 

The authors convincingly show that:

  1. LJV transmission primarily occurs through the fecal-oral route
  2. Venereal transmission appears to be secondary to the fecal-oral route, though further studies are required to establish this mode of transmission.
  3. LJV Infections retards development of the fly
  4. LJV infections reduced foraging activity.

These conclusions are well supported by statistical analyses.

Minor:

Line 25: the authors should clarify where invasion of the fly occurred.

Line 51: Iflaviviridae; I don't think this is a family of viruses; should be Flaviviridae; perhaps "I" refers to "Insect". 

Figure 1. Delete. 

Perhaps the authors could expand the discussion on the problems associate with slow kill rate and a realist assessment of producing a product based on LJV; such a discussion could include potential engineering of the virus to enhance kill rate, in addition to the SIT. 

Author Response

Reviewer 2:

Drosophila suzukii is emerging as a viable threat to the fruit and wine industries globally. This fruit fly species is difficult to control for several reasons, including resistance to insecticides and is location and foraging inside fruits.  The authors conducted routine and basic work to build upon a previous study that suggested the La Jolla virus (LJV) could be an alternative biocontrol agent for this pest. In this study the authors characterized the transmission and pathology of LJV by investigating the modes of transmission (airborne, venereal and oral, and fecal routes); feeding habits; and fecundity. 

The authors convincingly show that:

  1. LJV transmission primarily occurs through the fecal-oral route
  2. Venereal transmission appears to be secondary to the fecal-oral route, though further studies are required to establish this mode of transmission.
  3. LJV Infections retards development of the fly
  4. LJV infections reduced foraging activity.

We thank reviewer 2 for his positive response.

These conclusions are well supported by statistical analyses.

Minor:

Line 25: the authors should clarify where invasion of the fly occurred.

Thank you for your comment. The initial spread was in Hawaii in 1980 outside its endemic distribution, but in 2008, the fly simultaneously was detected in California and in Europe, Spain and Italy. We have added “in California and Europe” in the first paragraph (line 25).

Line 51: Iflaviridae; I don't think this is a family of viruses; should be Flaviviridae; perhaps "I" refers to "Insect". 

We do think that Iflaviridae is a family of virus, we refer to the ICTV Virus Taxonomy Profile: Iflaviridae by Valles and colleagues, 2017. The Abstract reads as follows:

Iflaviridae is a family of small non-enveloped viruses with monopartite, positive-stranded RNA genomes of approximately 9-11 kilobases. Viruses of all classified species infect arthropod hosts, with the majority infecting insects. Both beneficial and pest insects serve as hosts, and infections can be symptomless (Nilaparvatalugens honeydew virus 1) or cause developmental abnormalities (deformed wing virus), behavioural changes (sacbrood virus) and premature mortality (infectious flacherie virus). The host range has not been examined for most members. The most common route of infection for iflaviruses is the ingestion of virus-contaminated food sources. This is a summary of the International Committee on Taxonomy of Viruses (ICTV) Report on the taxonomy of the Iflaviridae, which is available at www.ictv.global/report/iflaviridae.

Please see also attached doi: https://doi.org/10.1099/jgv.0.000757

Figure 1. Delete. 

We deleted Figure 1 as requested.

Perhaps the authors could expand the discussion on the problems associate with slow kill rate and a realist assessment of producing a product based on LJV; such a discussion could include potential engineering of the virus to enhance kill rate, in addition to the SIT. 

We thank reviewer 2 for his valuable comment. We expand our discussion with “The mass-production of LJV for biological pest control would require the development of a cost-effective, large scale fermentation processes involving cell culture-based methods. Engineering could potentially enhance the speed and effectiveness of LJV's lethal capabilities. Nevertheless, the public generally disfavors genetically modified organisms (GMOs). As a result, approaches that utilize natural selection processes are more widely accepted as alternatives.” In line 396-401.

Round 2

Reviewer 1 Report

Comments and Suggestions for Authors

The authors have addressed all of my concerns with the original draft.